# Modeling the Copy Number of HSATII Repeats in Human Pericentromere

**DOI:** 10.3390/ijms26104751

**Published:** 2025-05-15

**Authors:** Puranjan Ghimire, Richard I. Joh

**Affiliations:** 1Department of Physics, Virginia Commonwealth University, Richmond, VA 23220, USA; 2Massey Comprehensive Cancer Center, Virginia Commonwealth University, Richmond, VA 23220, USA

**Keywords:** heterochromatin, pericentromere, copy number, human satellite, silencing, mathematical modeling, bifurcation

## Abstract

Tandemly repeated DNA fragments are major components of centromeres and pericentromeric heterochromatin, which is responsible for chromosomal stability and segregation. Recent evidence suggests that transcripts from these repeats play a key role in heterochromatin maintenance, and these repeats can be highly dynamic with various copy numbers. Here, we developed a mathematical model for human satellite repeats, which tracks the silenced and desilenced repeats, lncRNA, and copy number. Our model shows that chromatin factors for silencing and RNA stability can facilitate copy gain in satellites. Also, the system can be bistable, and cells with different copy numbers, silenced repeats with a small copy number, and desilenced repeats with a large copy number may coexist. To incorporate the cooperative methylation by neighboring repeats and the local chromatin environment, we also developed a spatial model where the local chromatin environment facilitates methylation locally. This model suggests that a local domain of silenced repeats may be an important feature of copy number regulation. Our models suggest that pericentromeric repeats are highly dynamic, and small changes in chromatin regulation can lead to large changes in satellite copy numbers.

## 1. Introduction

The centromere is a crucial chromosomal region, necessary for accurate chromosome segregation during cell division. The core centromere in humans is composed of α satellite DNA in tandem arrays of hundreds of kilobases to several megabases, and each α satellite unit is approximately 171 base pairs in length [1,2]. Surrounding the core centromere are the pericentromeric regions, often with diverse types of repetitive DNA, including satellite DNA, transposable elements, and segmental duplications. These sequences are more heterogeneous than the α-satellite DNA of the core centromere [3]. The pericentromeric regions can extend several megabases beyond the core centromere, with the length varying among different chromosomes and individuals [2]. These regions are vital for chromatin organization, kinetochore stability, and preserving the heterochromatin state necessary for centromere function [4].

Pericentromere is a crucial structural element of a chromosome, with its constitutive heterochromatic structure ensuring proper chromatin segregation through kinetochore formation during cell division and maintaining genome integrity [5,6]. Human pericentromeric DNA comprises various classes of tandemly repeated DNA, including β satellite, γ satellite, and satellites I, II, and III [7]. These pericentromeric repeats are rich in posttranslational modifications of histone tails such as H3K9me3, H4K20me3, and broad histone hypoacetylation [8,9,10]. In humans, the methyltransferases SUV39H1 and SUV39H2 are responsible for H3K9me3, a key marker for heterochromatin that is crucial for silencing pericentromeric repeats [11,12]. The H3K9me3 mark, deposited by SUV39H1 and SUV39H2, serves as a binding site for heterochromatin protein 1 (HP1), which is integral to chromatin compaction and the repression of pericentromeric repeats, thus maintaining genomic stability [13,14,15,16]. Furthermore, SUV39H1 and SUV39H2 have been found to function as both oncogenes and tumor suppressors in different types of cancer [17,18]. The homolog of SUV39H1/H2 in fission yeast (Clr4) is required for the silencing of pericentromeric repeats [19,20], and other chromatin factors determine the level of transcriptional silencing nonlinearly [21]. The deposition of H4K20me3 on pericentromeric heterochromatin is facilitated by SUV420H1 and SUV420H2, two SET-containing histone methyl transferases that localize to chromocenters via their interaction with HP1s [22,23].

Additionally, DNA methylation is an essential epigenetic factor that is responsible for heterochromatin formation [24,25], and its loss in the pericentromeric HSATII repeats has been reported in many types of cancers [26]. The DNA methyltransferases DNMT3A and DNMT3B are pivotal for the de novo methylation of pericentromeric repeats [27]. Conversely, TET enzymes (TET1, TET2, and TET3) are involved in DNA demethylation by converting 5mC to 5-hydroxymethylcytosine (5hmC) and other derivatives [28,29], and disruptions in their function can lead to aberrant demethylation in pericentromeric regions [30]. DNA hypomethylation in the HSATII regions of chromosomes 1 and 16 has been observed in various cancers [24,25], underscoring the importance of proper methylation in pericentromeric regions in carcinogenesis. This hypomethylation can occur very early in cancer and is strongly associated with tumor progression [25]. Additionally, overexpression of pericentromeric satellite transcripts has been reported in many cancer types [31,32], potentially resulting from the loss of DNA methylation in these regions [33]. Notably, DNA methylation of the pericentromeric region is associated with the HP1 proteins and H3K9me3 [34], indicating a complex interplay between DNA methylation and H3K9 methylation.

Recent discoveries show new roles of pericentromeric regions in our cells. Once thought to be transcriptionally silent, these areas may be actively transcribed during cell development to ensure proper chromosome formation [35,36]. Additionally, abnormal activation of a specific type of pericentromeric repeat, HSATII, has been linked to cancer [32,37]. The misregulation of HSATII RNA transcripts in cancer cells may lead to the dramatic expansion of HSATII through reverse transcription [38], which suggests a surprising link between HSATII RNA and the copy number gain of HSATII repeats. HSATII RNA is reverse transcribed into a DNA copy and inserted into the pericentromeric region, thus expanding the HSATII copy number [38]. Using reverse transcriptase inhibitors to reduce the reverse transcription of HSATII RNA in an in vivo tumor xenograft model leads to a decrease in HSATII DNA copy number gain and tumor growth [38]. In addition, *α* satellites are also associated with chromosomal instability and copy number alteration [38,39], and unique sets of ncRNAs are required for kinetochore assembly and cell-cycle progression [40]. These studies suggest the dynamic interplay between human pericentromere and ncRNAs.

The observed rise in copy number gain and DNA hypomethylation in cancer cells suggests a possible link between methylation and copy number variation in HSATII repeats. To explore this connection, we developed a mathematical model that considers RNA, methylated HSATII repeats, and total HSATII repeats as variables. The model uses simple nonlinear equations to track changes in these variables over time. Reverse transcription is included as a key process that introduces new copies into the pericentromere. Our model predicts that decreased methylation activity in HSATII repeats leads to an increase in copy number. It also indicates that the system can maintain both higher and lower steady states of copy number. Additionally, we propose a spatial model where repeats are arranged as monomers in a chain, and the state of a repeat in the next generation depends on its state in the previous generation. This model supports the ordinary differential equation (ODE) model’s result that decreased methylation activity in HSATII repeats leads to an increase in copy number.

## 2. Results

### 2.1. Mathematical Modeling of Gene Silencing at Pericentromeric Repeats

We have developed a mathematical model for gene silencing in HSATII repeats of the human pericentromere. The model keeps track of the number of HSATII transcripts, the number of methylated HSATII repeats (M), and the overall total number of HSATII repeats (CN). Unmethylated repeats can be estimated by the total number of repeats minus the number of methylated repeats. A schematic representation of the molecular pathway used to describe our mathematical model is shown in Figure 1. Even though pericentromeric repeats are silenced by both histone and DNA methylation, they are often highly correlated. Hence, in this model, we have assumed that HSATII repeats are either methylated (M), irrespective of H3K9 or DNA methylation, or unmethylated (U). RNA transcribed from unmethylated HSATII repeats can undergo reverse transcription [38], resulting in additional repeats. Both transcription and reverse transcription activities are reduced by methylation, which, in our model, is incorporated as being suppressed by the M repeat. HSATII copies can be lost due to the deletion via recombination [4,41]. The conversion of the unmethylated repeat (U) into the methylated one (M) can be either spontaneous or through neighbor-mediated local cooperative interactions tethered by HP1 or chromodomains of histone methyltransferases [42,43]. Similarly, HP1 can recruit DNMT3A and DNMT3B, which catalyze CpG methylation [27]. This neighbor-dependent interaction is represented in the model as the cooperative conversion of U to M. A feedback loop is generated as abundant RNAs can lead to many copies of repeats, and a greater number of repeats will transcribe more RNA. In this model, each repeat can be transcribed independently in the absence of methylated repeats.

### 2.2. HSATII Copy Gain with Reduced Methylation

To quantify the dynamic nature of the copy number of HSATII repeats in the human pericentromere, we found the steady states of our models, described by Equations (1)–(3). Figure 2 illustrates the ODE solutions when model parameters vary. The silencing is maintained by cooperative methylation and spontaneous methylation. Decreasing the cooperative methylation rate (*ϕ*) and the spontaneous methylation rate (*η*) leads to an increase in transcripts and the repeat copy number (Figure 2A,B). On the other hand, the methylated repeats can undergo demethylation, which can counterbalance the silencing by methylation. Increasing the demethylation rate destabilizes the heterochromatin, leading to more transcripts and higher copy numbers (Figure 2C). This suggests that stabilizing or promoting methylation increases methylation, which, in turn, leads to fewer transcripts and small copy numbers.

### 2.3. Bistability and Potential Coexistence of Higher and Lower Steady States of Copy Numbers

To understand how each model parameter affects the steady-state dynamics of the HSATII copy number, we estimated the steady state at a given parameter set and tracked the steady states as we varied one parameter. The resulting bifurcation diagram shows how the steady states change as a function of a model parameter. Figure 3A shows the bifurcation diagram of the cooperative methylation rate. As the cooperative methylation rate decreases, the fraction of methylated copies decreases, which, in turn, increases the number of transcripts and HSATII copy numbers. The bifurcation diagram also shows that the system can be bistable with two stable steady states, with a high and low fraction of M repeats. The silenced branch, represented by the red curve, often exhibits a high fraction of methylated repeats, low transcripts, and a small copy number. The desilenced branch, represented by the blue curve, is associated with low methylation, high transcripts, and a high copy number. The desilenced state represents that some of the repeats are transcriptionally active, which can produce lncRNAs. Identification of the silenced and desilenced branch is based on the relative fraction of methylated repeats, and the branch with lower methylation is marked as the desilenced branch. Hysteresis may be observed when the cooperative methylation rate varies continuously, and where the silenced and desilenced branches meet represents the threshold value for bistability. The increase in the RNA degradation rate (*δ*), on the other hand, can destabilize methylation and silencing among repeats (Figure 3B). The bifurcation diagram for other parameters is shown in Appendix A. The presence of a bistable regime means that it is possible that a small change in the parameter, like reaction rates affecting RNA, silencing, and recombination, can result in a large change in the steady states [44]. Such large changes in steady states can be observed when parameters change between monostability and bistability, and it also depends on the initial conditions. For example, if the system was silenced, the increase in *δ* (around *δ* = 0.022 in Figure 3B) can lead to monostability, where all solutions are desilenced regardless of initial conditions. In addition, some diseased cells may already be in bistable states, where a fraction of identical cells show large repeat copy numbers. For example, the repeat expansion observed in solid tumors may be due to small changes in the underlying reaction kinetics leading to bistability, unlike the monostably silenced WT condition.

### 2.4. Analysis of Steady States with Nullclines

The existence of bistability in the system for a range of parameter values is also supported by the nullcline analysis. We assumed a quasi-steady-state approximation (QSSA), where RNA reaches a steady state faster than other variables, to obtain the equations for the M nullcline and the CN nullcline (see Methods for details). In general, transcription and RNA degradation work on a time scale of minutes, whereas histone modifications or copy number changes occur on a much longer time scale. This can reduce the full three-dimensional system into two dimensions, which allows the visual representation of steady states at different parameter values. Intersections of nullclines represent the steady states. When there is one intersection, the system is monostable. However, nullclines can intersect at three points, and the system becomes bistable through a saddle-node bifurcation (two stable and one unstable steady states). The reference parameter set, corresponding to the WT condition, is monostable with one steady state (intersection of solid lines). Figure 4A shows how the M and CN nullclines change for different values of the cooperative methylation rate (ϕ). CN nullcline is independent of ϕ (Equation (6)) and does not change with a changing ϕ, while the M nullcline shifts upward with a decrease in ϕ (Figure 4A). There is a range of ϕ that can lead to bistability. On the other hand, the CN nullcline moves upward with an increase in the reverse transcription rate (*σ*) (Figure 4B). Increasing σ leads to bistability and further increase in σ leads to monostability with a large number of methylated repeats. The nullcline analysis for all the parameters is shown in Appendix A. Except for the half maximum number of methylated repeat saturation (*κ*_4_), bistability is observed for all the parameters. Figure 4C shows the list of parameters whose increase leads to the copy gain or loss. Our analysis shows that bistability can arise from different combinations of reaction rates and is not restricted to a specific parameter set.

### 2.5. Parameters for Bistability

To identify the range of parameters that leads to bistable, silenced, and desilenced states, we performed the sensitivity analysis by tracking the steady states while systematically changing one parameter at a time. Different initial conditions were used for each parameter set, and we determined if a parameter set leads to desilenced monostability, silenced monostability, or bistability. Figure 5A shows the summary of the dynamical state as we vary the indicated parameters. All parameters except for *κ*_4_ (half maximum M for the repression of a recombination) exhibit a distinct range of three possible states. Parameters *α*, *κ*_1_, *κ*_3_, *ϕ*, and *σ* lead to silencing with high values, while parameters *δ*, *γ*, *κ*_2_, and *μ* lead to desilencing with high values. In addition, we varied two parameters simultaneously. The cooperative methylation rate (*ϕ*) and the demethylation rate (*μ*) act antagonistically (Figure 5B), and the transcription rate (*α*) and the reverse transcription rate (*σ*) act synergistically for silencing (Figure 5C). Mutations affecting multiple silencing pathways may shift the steady states, or they may be compensated by other reactions.

### 2.6. Spatial Model Predicts the Copy Number Expansion with Local Cooperative Methylation

Although our ODE-based model assumes that there is cooperative methylation depending on the number of methylated repeats, the cooperative methylation happens through local interactions, which depend on the chromatin environment of neighboring repeats only. Histone methyltransferases for H3K9me, like SUV39H1/H2 or Clr4, have chromodomains, which can bind to methylated histones and are essential for maintaining silencing [19,43]. To incorporate such a cis interaction driven by the silencing of neighboring repeats, we also developed a spatial model where each repeat is modeled as a 1D chain of repeat monomers (see Methods and Figure 6A). Each repeat is either methylated (U) or unmethylated (M), and the methylation can be spontaneous or cooperative. Cooperative methylation depends on the presence of nearby methylated repeats within two repeats. The model is Markovian, and the next state depends only on the current state of all repeats. The number of Us and Ms as well as the local cooperative interactions and the number of lncRNAs determine the next-generation state. Figure 6A represents all the interactions, and the repeat copy number can increase or decrease. The fraction of methylated repeats is high when the methylation rate is high, and all repeats lose methylation if these rates are low (Figure 6B). This spatial model can recapitulate the behavior of the ODE model, where the copy number and methylated repeats are modulated by the same model parameters.

In addition, the model behavior is more complex than the ODE model, as the cooperative methylation is only dependent on the neighboring states. Figure 6C,D show a representative simulation where the majority of repeats are methylated. The copy number of repeats can fluctuate between 14,000 and 17,000 copies. The fraction of M is typically around 68%, but a low or high M fraction (around 43% or 77%) can lead to copy gain or loss. The accumulation of U can weaken the cooperative methylation, which, in turn, increases the copy number around generation 40. In addition, if the average domain size of M (consecutive M with up to 20% U in between) increases, then the cooperative methylation is favored, which results in the loss of copies around generation 60. This model can capture that repeats can be highly dynamic, and small changes in local states can lead to copy number changes.

## 3. Discussion

We developed a mathematical model of silenced and desilenced human satellite repeats to make testable predictions on how different silencing-associated reactions affect the repeat copy number. Centromeric and pericentromeric human satellites form a complex structure, which varies in its composition among different chromosomes, including HSAT1A, HSAT1B, HSATII, and HSATIII [45,46]. Many of these repeats are often difficult to study with large genomic studies, and this is due to the identical nature of such repeats and potential repetitive DNA amplification during the sample preparation. We used a mathematical approach to bypass these issues with simplified biological interactions, which may provide insights into the key processes associated with the copy number changes in human satellites. Although the expansion of human satellite repeats is well documented in various types of cancer [32,47,48], still, there are only a few quantitative modeling approaches [49,50]. This is in part due to the uncharacterized cofactors for silencing-associated enzymes, which act in a highly redundant manner in higher eukaryotes. Our model can make testable predictions in terms of how underlying silencing-mediated processes affect the satellite copy number.

HSATII is associated with tumor progression, but our understanding of the regulation of human satellite RNAs is an active area of research. The expansion of HSATII via the endogenous HSATII transcripts was reported not only in cancer cells but also in herpes infection [51], which might be via a noncanonical ATM-regulated DNA damage response [52]. Another study has shown that HSATII foci can form cancer-specific nuclear bodies and sequester PRC1 and MeCP2, acting as a molecular sponge [41]. Our model suggests that the increase in copy number alone can support desilenced repeats due to the inherent bistability, and an occasional copy gain can lead to changes in the transcript and chromatin levels. A recent study also suggests that HSATII may be up-regulated by herpesviruses [51], suggesting that its stability may be important to other human diseases. Other than HSATII, HSATIII copy numbers can vary with aging and schizophrenia [53,54]. In addition, HSAT1A was largely disregarded in genomic and transcriptional studies with an AT-rich fraction, but size-varying polyadenylated HSAT1A transcripts were reported [55]. These studies suggest that the biology of repetitive elements may be associated with different human diseases.

Our model is minimal, and one of the main limitations is the detailed chromatin state into two groups, only desilenced and silenced, due to the colocalized and redundant silencing machinery. Heterochromatin, including the pericentromeric region, is characterized by H3K9me2/me3 [56], and DNA methylation is also highly correlated with histone methylation [57,58]. This may be due to the physical interaction between SUV39HA and DNAM1/3A [34,59,60]. Our previous model on fission yeast also included the acetylated H3K9 [21], whose qualitative behavior was consistent with the simpler U/M model. It is possible that under certain conditions, the coregulation between H3K9me and DNA methylation is abolished, but in most cases, they act redundantly to ensure the silencing of heterochromatin regions. Our model works with a small number of assumptions and may work with other repetitive DNA elements. Other short interspersed elements (SINEs), including Alu and long interspersed elements (LINEs), occupy a large fraction of the genome, close to 50% of the total genome [61,62,63,64]. LINE and SINE are often more dynamic with different tissue-by-tissue expressions [65]. Such repetitive elements work similarly to the human satellites, which are often silenced by DNA methylation with H3K9me and H3K27me. Our spatial model suggests that the domain size of methylated repeats may play a role in copy number regulation, and modeling each chromatin level modification of H3K9me, H3K27me, and DNA methylation separately would be able to determine the role of heterochromatin boundary. A multi-omics approach would be necessary to capture the dynamics at the level of lncRNA, chromatin, DNA methylation, and the satellite copy number. In the heterochromatin, these processes are highly correlated and redundant; it would be interesting to see if the dynamics of different markers are delineated in weakly silenced regions like a small tandem array.

Other studies have shown that satellite DNA sequences may act as an active regulatory component. Different cellular stresses can induce the overexpression of transcripts from such repeats, and stress includes heat, oxidate stress, osmotic stress, and DNA damage responses [66]. Upregulation of satellite transcripts is not restricted to mammals and is observed in insects, as well [67,68]. This suggests that the regulation of repetitive DNA elements may share conserved biological functions among different species, and promoting repeat copy gain may be advantageous to cells under certain environmental conditions. In addition, lncRNAs with very large copy numbers may act as a robust stress response mechanism with a high signal-to-noise ratio.

## 4. Materials and Methods

### 4.1. ODE-Based Mathematical Modeling

Our quantitative model keeps track of the number of lncRNAs associated with HSATII repeats (number of transcripts), the methylated HSATII repeat copy number, and the total HSATII repeat copy number (including both methylated and unmethylated forms), using three first-order nonlinear differential equations, which are denoted as x_1_, x_2_, and x_3_, respectively. Three ODEs describe the time evolution of x_1_, x_2_, and x_3_ (Equations (1)–(3)),(1)$$\dot{x_{1}}=\frac{\alpha x_{3}}{1+\frac{x_{2}}{\kappa_{1}}}-\delta x_{1},$$(2)$$\dot{x_{2}}=\frac{\frac{\phi x_{2}}{\kappa_{2}}x_{2}(x_{3}-x_{2})}{1+\frac{x_{2}}{\kappa_{2}}}+\eta \left(x_{3}-x_{2}\right)-\mu x_{2},$$(3)$$\dot{x_{3}}=\frac{\sigma x_{1}}{1+\frac{x_{2}}{\kappa_{3}}}-\frac{\gamma x_{3}^{2}}{1+\frac{x_{2}}{\kappa_{4}}}.$$

The first term of Equation (1) represents that the RNA transcription is proportional to the total number of repeats at a rate α per copy, which is repressed by methylation. The second term of Equation (1) indicates the RNA degradation at a rate δ. Methylation can occur through cooperative methylation, described by the first term of Equation (2). Other terms of Equation (2) represent spontaneous methylation (rate η) and demethylation (rate μ). A new repeat can be added through insertion after reverse transcription (rate σ), and a repeat can be lost via recombination (rate γ). An additional description of all parameters can be found in Appendix A. Here, our goal is to develop a simple mathematical model where methylated repeats are a combination of multiple silencing marks like DNA methylation, H3K9me, and H3K27me, and unmethylated repeats can be either unmethylated or acetylated.

### 4.2. Solutions of ODE

The system of Equations (1)–(3) was solved in MATLAB (R2023a) [69] using the ode23 solver for seven different initial counts of RNA, methylated repeats (M), and copy numbers (CN) of HSATII repeats. The parameter values used for wild-type cells are provided in Appendix A, and the seven initial conditions used in the model are listed in Appendix A.

### 4.3. Bifurcation Diagram

To construct a bifurcation diagram of steady states with respect to the parameter values, Equations (1)–(3) were solved for seven initial conditions until steady states were reached for each indicated parameter set. The resulting seven steady states were grouped based on the fraction of methylated repeats as silenced and desilenced states, using the k-means algorithm with k = 2 [70].

### 4.4. Quasi-Steady-State Approximation (QSSA)

To determine if the bistability can be achieved analytically, we assumed that RNA saturates much faster than the methylated repeat (M) and total repeat (CN), meaning RNA is in equilibrium with respect to M and CN at any given time. The time scales of these dynamics are different, with minutes for the transcription and RNA lifetime, hours to days for the methylation, and several days for the copy number changes. Therefore, we can assume that the dynamics of RNA happen while M and CN hold almost constant. With this QSSA, we can find the steady-state value of RNA by setting Equation (1) = 0 and rearranging for x_1_, which leads to(4)$$x_{1}^{*}=\frac{\alpha x_{3}}{\delta (1+\frac{x_{2}}{\kappa_{1}})}.$$

Then, the M nullcline can be found by setting Equation (2) = 0 as(5)$$x_{3}=\frac{\frac{\phi \frac{x_{2}}{\kappa_{2}}x_{2}^{2}\;}{1+\frac{x_{2}}{\kappa_{2}}}+\eta x_{2}+\mu x_{2}}{\frac{\phi \frac{x_{2}}{\kappa_{2}}x_{2}}{1+\frac{x_{2}}{\kappa_{2}}}+\eta }.$$

Then, the CN nullcline can be found by using Equation (4) and setting Equation (3) = 0 as(6)$$x_{3}=\frac{\frac{\sigma \alpha }{\gamma \delta }(1+\frac{x_{2}}{\kappa_{4}})}{\left(1+\frac{x_{2}}{\kappa_{1}})(1+\frac{x_{2}}{\kappa_{3}}\right)}\;\mathrm{or}\;x_{3}=0.$$

The intersections of Equations (5) and (6) represent the system’s steady states. If there is one intersection, the system is monostable. It is possible to have multiple steady states, including both stable and unstable steady states (Figure 4).

### 4.5. Sensitivity Analysis

To assess the sensitivity of one parameter, the parameter was systematically varied while the other parameters were kept at their reference values. We tracked the steady states from different initial conditions and the presence/absence of desilenced and silenced states. This result is illustrated in the bar graph (Figure 5A). Each bar shows the range of values where the system is in the silenced, desilenced, or bistable state for a specific parameter.

In addition, we also varied selected pairs of parameters simultaneously while fixing all other parameters. Figure 5B,C represent the representative example where the color represents the silenced, desilenced, or bistable state. The figure shows three distinct regions of stability, with clear transitions from one state to another, and vice versa.

### 4.6. Spatial Model of Repeats

In the spatial model, each repeat is either in an unmethylated (U) or methylated (M) state and is arranged as monomers in a polymer chain (see Figure 6). U can turn into M via spontaneous (rate *η* = 0.01) or cooperative (rate *ϕ* = 0.2) methylation. Cooperative methylation can happen only if there is M in the vicinity of U (within two repeats). Additionally, M can convert to U through demethylation (rate *μ* = 0.1). A new repeat can be inserted at U via reverse transcription, and the transcript level is determined by the total U and M. U can be deleted by recombination with other Us. The state of U/M and insertion/deletion events are stochastic, determined by a random number, and the current state determines the next-generation state. The transcript level is determined as $n\left(transcript\right)=\frac{\alpha \times CN}{\delta (1+\frac{nM}{\kappa })}$, which is proportional to the total copy number and discounted by methylation. The insertion was set by $\sigma \times \frac{n\left(transcript\right)}{1+\frac{nM}{\;\kappa_{3}}\;}$, which represents the reverse transcription repressed by methylation. The deletion was set by $\gamma \times \frac{CN^{2}}{1+\frac{nM}{\kappa_{4}}}$, which represents the recombination between two repeats. Here, nM and CN denote the number of methylated repeats and the total copy number, respectively.

## Figures and Tables

**Figure 1 ijms-26-04751-f001:**
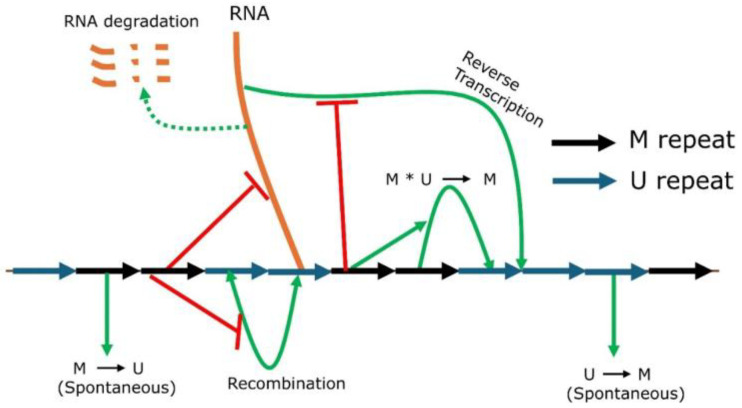
Schematic diagram for mathematical modeling of HSATII repeat silencing in humans. HSATII repeats are arranged linearly as blue (unmethylated U) or black arrows (methylated M) in the pericentromere. RNA transcription is suppressed by methylated repeats. Methylation can occur spontaneously or be cooperatively (M * U → M) facilitated by neighboring M, and demethylation occurs only spontaneously. Recombination and reverse transcription are suppressed by methylation.

**Figure 2 ijms-26-04751-f002:**
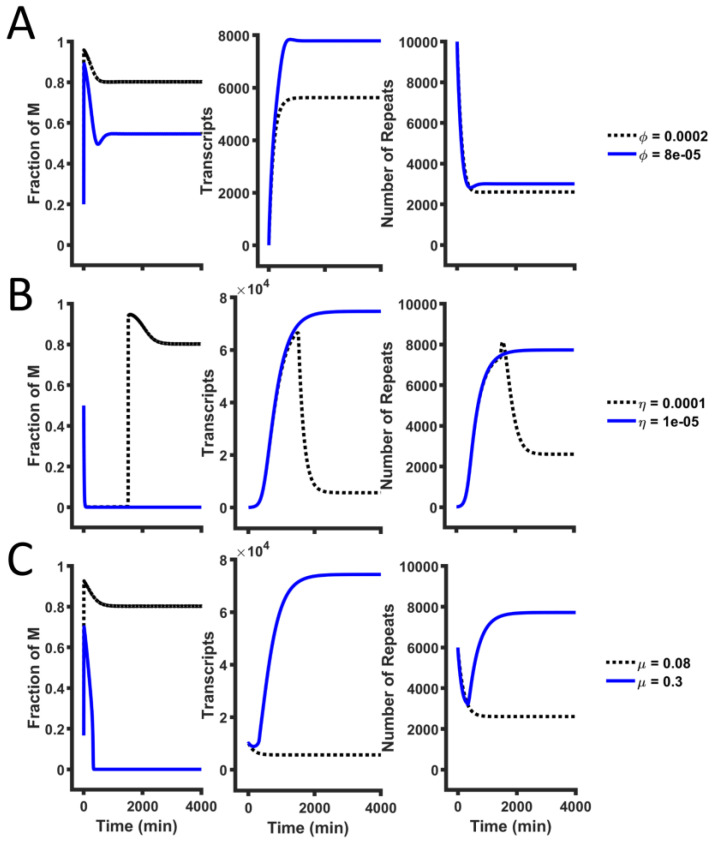
Strong methylation leads to few HSATII repeats. The solution of our mathematical models (Equations (1)–(3)) while varying (**A**) the cooperative methylation rate (*ϕ*), (**B**) the spontaneous methylation rate (*η*), and (**C**) the demethylation rate (*μ*). The black dotted lines represent the reference parameter set (WT), and the blue lines represent cases where silencing is weakened by a change in one parameter.

**Figure 3 ijms-26-04751-f003:**
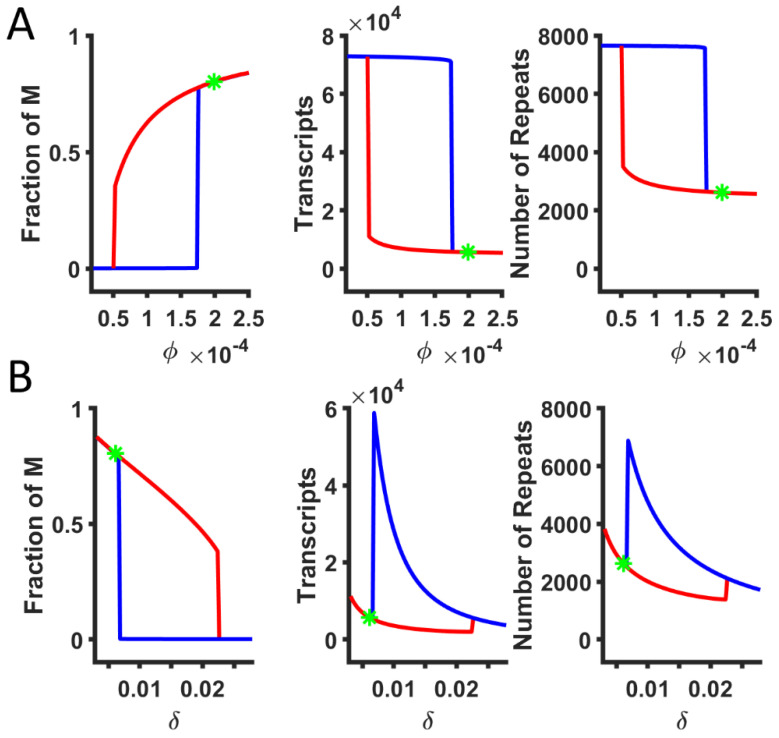
Bifurcation diagrams of steady states while varying (**A**) the cooperative methylation rate (*ϕ*) and (**B**) the RNA degradation rate (*δ*). The green star represents the reference value (WT) of the parameter, which is monostably silenced. The red and blue lines indicate the silenced and desilenced branches, respectively.

**Figure 4 ijms-26-04751-f004:**
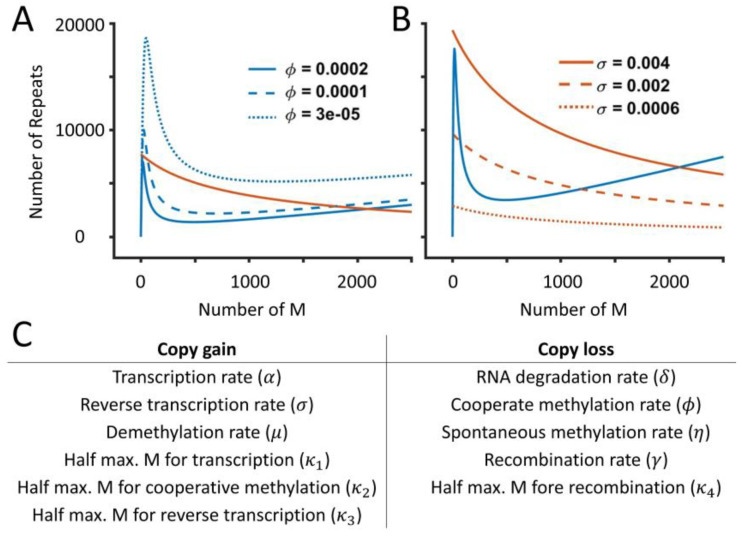
Nullcline analysis shows the existence of bistability while varying (**A**) the cooperative methylation rate (*ϕ*) and (**B**) the reverse transcription rate (*σ*). Nullclines are obtained by the QSSA, and the orange and blue curves represent the CN and M nullclines, respectively. (**C**) A list of parameters whose increase leads to a copy number gain or loss.

**Figure 5 ijms-26-04751-f005:**
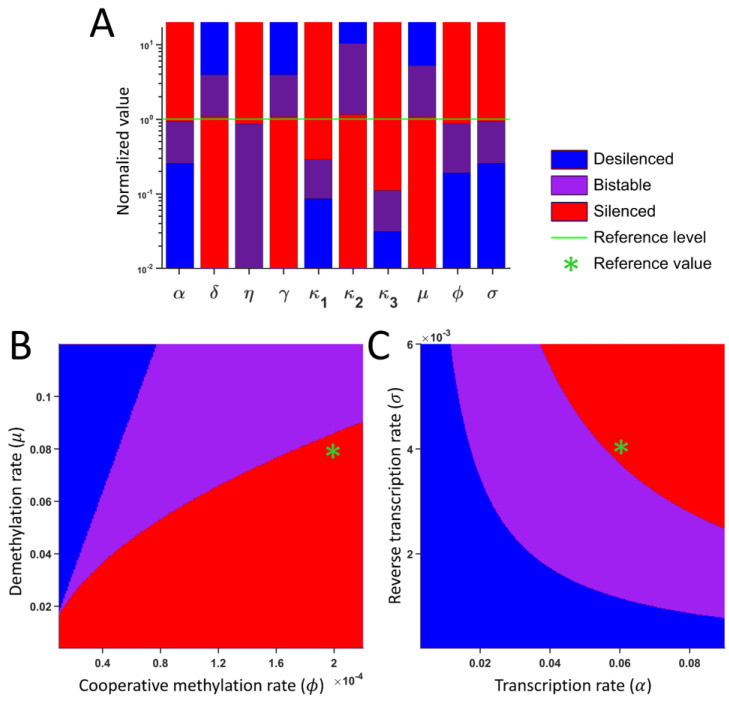
Sensitivity analysis: (**A**) Qualitative change in the system behavior as the indicated parameter is varied. The green line (normalized value 1) represents the reference value. The blue, red, and purple colors represent the desilenced, silenced, and bistable states as the indicated parameters change with respect to the reference value, respectively. The normalized value represents the parameter value divided by the reference value. (**B**,**C**) Qualitative change in the system behavior while varying (**B**) the cooperative methylation rate (*ϕ*) and the demethylation rate (*μ*), and (**C**) the transcription rate (*α*) and the reverse transcription rate (*σ*).

**Figure 6 ijms-26-04751-f006:**
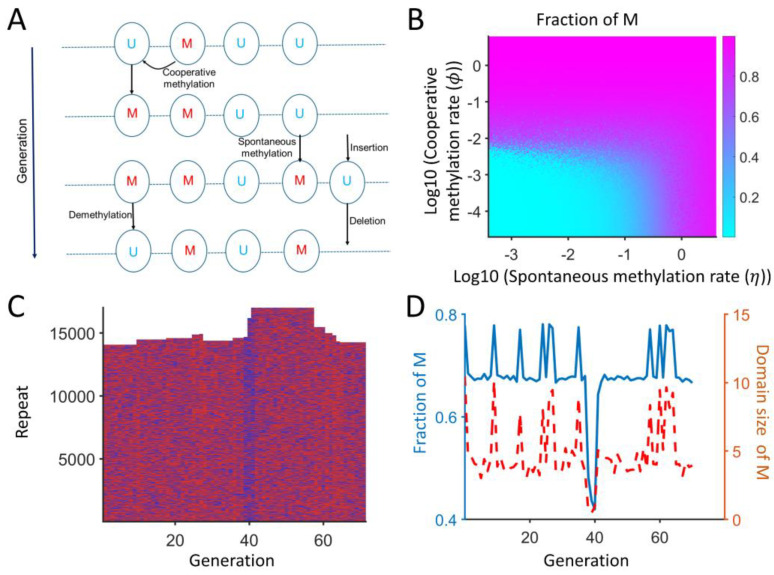
Schematic diagram for the spatial model and its behaviors. (**A**) Each repeat is spatially embedded in a 1D chain of repeat monomers with length CN, and its state is determined by demethylation, spontaneous (in trans), and cooperative (in cis) methylation. Each repeat can be methylated (M) or unmethylated (U). A new repeat can be added through reverse transcription, and a repeat can be deleted by recombination. (**B**) The fraction of methylated repeats when the spontaneous and cooperative methylation rates are varied. (**C**) A representative evolution of repeats. The red and blue colors represent M and U, respectively. (**D**) The fraction of M and the domain size of M over generations.

## Data Availability

Data associated with the findings of this study are available from the corresponding author (R.I.J.) upon request. Codes will be available at www.johlab.com.

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
