# Peer review of "Modeling the Copy Number of HSATII Repeats in Human Pericentromere"

_ijms, 2025, doi:10.3390/ijms26104751_

Round 1
Reviewer 1 Report
Comments and Suggestions for Authors
This paper presents a novel and interesting mathematical model which predicts copy number variations in pericentromeric heterochromatin.
The findings are well presented, although I recommend addressing the following points to strengthen the reported findings.
Main Points:
1. I would suggest to the authors to expand more on the relevance of their matemathical model in the discussion. Right now it is now clear which is the relevance of their mathematical model and how other researches might take advantage of using it for their studies. Also, the first sentence of the discussion is not clear enough. Was the model developed to predict, to study or to better understand copy number variations in pericentromeric heterochromatin?
2. Expand more on the results from Figure 2.
Minor points:
-line 51-52: homolog of SUV39H1/H2, in which system?
-line 73: recent discoveries are discovering - rephrase
-line 98: ODE, please report the acronym in full for the first time
Author Response
1. I would suggest to the authors to expand more on the relevance of their matemathical model in the discussion. Right now it is now clear which is the relevance of their mathematical model and how other researches might take advantage of using it for their studies. Also, the first sentence of the discussion is not clear enough. Was the model developed to predict, to study or to better understand copy number variations in pericentromeric heterochromatin?
> Thank you for your suggestion. We have expanded the role of quantitative modeling in Discussions.
2. Expand more on the results from Figure 2.
> Thanks for the suggestions. We have modified the text in Section 2.2 as well as updating Figure 2.
Minor points:
-line 51-52: homolog of SUV39H1/H2, in which system?
> We have included the information (homolog in fission yeast (Clr4) in the main text.
-line 73: recent discoveries are discovering – rephrase
> We have rephrased this sentence.
-line 98: ODE, please report the acronym in full for the first time
> We included the full term as suggested.
Reviewer 2 Report
Comments and Suggestions for Authors
The authors provide a mathematical model to address the dynamics of satellite repeats in a human chromosome. With simple and experimentally based assumptions they suggest 11 parameters that affect how the copy number, the number of methylated repeats and the number of RNA transcripts from unmethylated repeats change with time. This is an insightful resource to the community not only for its applications on medicine, but to understand evolution of genomes, since many plant species in the same genus have been described by their different composition of repeats. Besides minor revisions that are needed to improve clarity, to prove the impact of this research, I also suggest an introduction on the existence of previous similar models, and a discussion on the limitations of the present model and how the expected consequences of changing the analysed parameters can be verified experimentally, for example, that the increase in the methylation rate leads to more repeats.
Suggested revisions
Line 52 – which homolog, from which species?
Line 73 – discovery redundancy
Line 287 – “track of the number OF THE lncRNAs associated with HSATII repeats, OF THE methylated HSATII repeat copy number, and OF THE total HSATII repeat copy number”
Line 288 – “number of lncRNA associated with HSATII (number of transcripts)”
Line 309 – eqs. were solved
Section 2.2 and figure 2 - not clear if f, h, and m are varied independently from one another while maintaining the others at the reference value. If they are, and the blue line is always the wild type condition (which should be mentioned explicitly), the blue line should always be the same for A, B and C. Initial conditions also seem to be different in B, it would be good to indicate the set of initial conditions used. A and C graphics also seem exactly the same, is it really so that the changes in cooperative methylation and demethylation had the exact same effect, please mention if this is case, or was it just a mistake in the figures?
Line 140 – “This suggests that the stabilizing”
Section 2.3 figure 3 - From the graphics of methylation fraction, the k means algorithm seems to have grouped the 7 different simulations at either 0 methylation fraction or higher, non-zero methylation fraction, characterizing the desilenced or silenced branches, respectively. But in the discussion, the desilenced branch is described as low methylation fraction, instead of zero. This seems a euphemism of the authors which shades the description of the results. Please describe it precisely as zero methylation fraction, or mention the exact range of methylation fraction values for the desilenced branch (if it is not exactly 0 as it appears in the graphics), or explain why it is preferable to describe it as low methylation fraction even if the result indicates exactly 0.
Line 160 – “The presence of a bistable regime means that it is possible that a small change in the parameter, reaction rates affecting RNA, silencing, and recombination, can result in a large change in the steady states”. Is this true regardless of the initial conditions? Please shortly mention the effects of different initial conditions in this section.
Line 318 – equation 4 is not needed to find the M nullcline.
Line 319 – “eq. 3 = 0” in bold.
Line 320 – This sentence is unclear, please rephrase it. Maybe like this: The intersections of Eqs. 5 and 6 represent the system’s steady states. Multiple steady states may exist, including both stable and unstable ones.
Figure 4 is missing indication for subpanel B
Line 191 – Please add to the last statement of this section, if the set of parameters for wild type cells is bistable or not.
Section 4.5 – Were different simulations with different initial conditions also used for the sensitivity analysis?
Figure 5A is hard to understand. The meaning of the bars, present in the methods section 4.5, should be in the legend. It is not clear why the bars have different heights. The meaning of the arrows is also missing.
Line 219 – “the cooperative methylation is happens through local interactions, which depend on”
Figure 6A – Please describe the chain model better in the legend. Perhaps adding to the to the sentence in line 212 “each repeat is spatially embedded in a 1D polymer chain of length CN, …”
Author Response
The authors provide a mathematical model to address the dynamics of satellite repeats in a human chromosome. With simple and experimentally based assumptions they suggest 11 parameters that affect how the copy number, the number of methylated repeats and the number of RNA transcripts from unmethylated repeats change with time. This is an insightful resource to the community not only for its applications on medicine, but to understand evolution of genomes, since many plant species in the same genus have been described by their different composition of repeats. Besides minor revisions that are needed to improve clarity, to prove the impact of this research, I also suggest an introduction on the existence of previous similar models, and a discussion on the limitations of the present model and how the expected consequences of changing the analysed parameters can be verified experimentally, for example, that the increase in the methylation rate leads to more repeats.
> We would like to thank the reviewer for your valuable comments. In addition to answering your specific comments, we have expanded the discussions to 1) present other models, 2) discuss the model limitations, and 3) experimental validations.
Suggested revisions
Line 52 – which homolog, from which species?
> Clr4 is the SUV39H1/H2 homolog in fission yeast (S. pombe). We added this information in the main text.
Line 73 – discovery redundancy
> We have changed the text as “Recent discoveries show new roles of pericentromeric regions”.
Line 287 – “track of the number OF THE lncRNAs associated with HSATII repeats, OF THE methylated HSATII repeat copy number, and OF THE total HSATII repeat copy number”
Line 288 – “number of lncRNA associated with HSATII (number of transcripts)”.
>We have modified the text as suggested. The copy number already means the number of repeats.
Section 2.2 and figure 2 - not clear if f, h, and m are varied independently from one another while maintaining the others at the reference value. If they are, and the blue line is always the wild type condition (which should be mentioned explicitly), the blue line should always be the same for A, B and C. Initial conditions also seem to be different in B, it would be good to indicate the set of initial conditions used. A and C graphics also seem exactly the same, is it really so that the changes in cooperative methylation and demethylation had the exact same effect, please mention if this is case, or was it just a mistake in the figures?
>Thanks for pointing it out. Yes, blue lines are from the reference set, and different initial conditions were used for Fig. 2B. We have modified the Figure 2. In the new figure, black lines represent the WT case, where blue lines are cases were silencing is weakened. Figure 2 legend was modified as well. Results of previous Fig 2A and 2C shows about 5% deviation in the results, and we changed the parameter so that they look different in the revised figure.
Line 140 – “This suggests that the stabilizing”
>We have modified the text as suggested.
Section 2.3 figure 3 - From the graphics of methylation fraction, the k means algorithm seems to have grouped the 7 different simulations at either 0 methylation fraction or higher, non-zero methylation fraction, characterizing the desilenced or silenced branches, respectively. But in the discussion, the desilenced branch is described as low methylation fraction, instead of zero. This seems a euphemism of the authors which shades the description of the results. Please describe it precisely as zero methylation fraction, or mention the exact range of methylation fraction values for the desilenced branch (if it is not exactly 0 as it appears in the graphics), or explain why it is preferable to describe it as low methylation fraction even if the result indicates exactly 0.
>We would like to clarify that the notion of silenced and desilenced branch is relative. K-means algorithm clusters the different simulations into two groups, where the groups with lower and higher methylation is terms desilenced and silenced, respectively. The fraction of methylated repeats is not zero (unless we set some parameters zero), and desilenced repeats means some of repeats are transcriptionally active (not necessarily all repeats). We added this information in the main text.
Line 160 – “The presence of a bistable regime means that it is possible that a small change in the parameter, reaction rates affecting RNA, silencing, and recombination, can result in a large change in the steady states”. Is this true regardless of the initial conditions? Please shortly mention the effects of different initial conditions in this section.
>You are correct that such changes depend on the initial conditions and how parameters change. The largest qualitative changes are expected when the parameter sets change between monostability and bistability. We have modified the main text to reflect this.
Line 318 – equation 4 is not needed to find the M nullcline.
>Thanks for pointing this out. We have modified the text so that M nullcline is directly from Eq.2=0.
Line 319 – “eq. 3 = 0” in bold.
>We have modified the text as suggested.
Line 320 – This sentence is unclear, please rephrase it. Maybe like this: The intersections of Eqs. 5 and 6 represent the system’s steady states. Multiple steady states may exist, including both stable and unstable ones.
>Thanks for the suggestion. We modified the text as “The intersections of Eqs. 5-6 represent the system’s steady states. If there is one intersection, the system is monostable. It is possible to have multiple steady states, including both stable and unstable steady states (Fig. 4).”
Figure 4 is missing indication for subpanel B
>Thank you for pointing this out. We have updated the Figure 4.
Line 191 – Please add to the last statement of this section, if the set of parameters for wild type cells is bistable or not.
>We have clarified that the WT parameter is monostable in the Figure 3 legend and Section 2.4.
Section 4.5 – Were different simulations with different initial conditions also used for the sensitivity analysis?
>You are correct, and we included this information in Sections 2.5 and 4.5.
Figure 5A is hard to understand. The meaning of the bars, present in the methods section 4.5, should be in the legend. It is not clear why the bars have different heights. The meaning of the arrows is also missing.
>To improve the readability, we have modified the Figure 5A while removing the arrows completely. The color represents the state (desilenced, silenced, or bistable) at the normalized value (reference value is 1). Also, we have modified the color scheme to be consistent from Fig 2-6, where red is silenced (high methylation), and blue is desilenced (low methylation).
Line 219 – “the cooperative methylation is happens through local interactions, which depend on”
> We have modified the text as suggested.
Figure 6A – Please describe the chain model better in the legend. Perhaps adding to the to the sentence in line 212 “each repeat is spatially embedded in a 1D polymer chain of length CN, …”
>We have expanded Fig. 6 legend as suggested.
Reviewer 3 Report
Comments and Suggestions for Authors
In this manuscript, Ghimire and Joh developed a mathematical model and used it to illustrate the dynamic influence of methylation stages (methylated versus unmethylated) and the copy number of HSATII repeats. They proposed a bistable model that represents the interplay between the methylation states and the copy number: methylated repeats are coupled with a small copy number, whereas unmethylated repeats tend to be associated with a large copy number.
Overall, the proposed model is well described, and this manuscript is well written. Some concerns that require further clarification by the authors have arisen after reading this work.
Major concerns
1. I understand the logic that in the first attempt to develop the model, the authors considered both histone and DNA methylation as the same kind. Indeed, even though biologically lots of crosstalk can manifest between them, they remain fundamentally different. Have the authors run the model in the separation between DNA and histone methylation?
In addition, it is recommended that authors elaborate on more details regarding histone and DNA methylation on HSATII repeats in the Discussion.
2. Based on the simulation results shown in Figure 3, the authors claimed the presence of bistability in this model. The two-state model is often characterized by a clear threshold response with deterministic bistability; however, this characteristic does not seem to be obvious, as illustrated in Figure 3A. In other words, I found that hysteresis does not clearly appear in this model, making me wonder whether or not the relation between methylated and unmethylated would be a transient without a clear threshold? Please comment on it.
3. In the Materials and Methods, 4.4. QSSA (p. 10), it is not clear to me why the authors consider that RNA is in equilibrium with respect to M and CN at any given time if they assume that RNA saturates much faster than M and CN. Please comment on it.
4. In the Introduction (line 97, p. 3), the authors claimed that the state of a repeat in the next generation depends on its state in the previous generation. However, the simulation results performed in the spatial model do not seem to support this conclusion, given that, overall, methylated repeats are dominant across testing generations, except generation 40. If the probability of conversion between methylated and unmethylated stages is stochastically determined, as mentioned in the Materials and Methods (then I assume it is a random walk modeling), shouldn’t each state (or generation) be independent? Can the authors explain more about the dependency mentioned between two generations?
5. In the Materials and Methods, 4.6. (p. 33), is it true that U can convert to M through “demethylation’? Or “methylation”?
Minor concerns
1. Figure 2: It is suggested that the authors keep the same color codes for representing the parameters possessing high or low values, i.e., the color codes used in panel C are opposite from panels A and B.
Author Response
Major concerns
1. I understand the logic that in the first attempt to develop the model, the authors considered both histone and DNA methylation as the same kind. Indeed, even though biologically lots of crosstalk can manifest between them, they remain fundamentally different. Have the authors run the model in the separation between DNA and histone methylation?
In addition, it is recommended that authors elaborate on more details regarding histone and DNA methylation on HSATII repeats in the Discussion.
> Thanks for your comments, and we agree that DNA and histone methylation is fundamentally different processes. However, especially in the typical heterochromatin regions, they are highly correlated, and several studies have shown that SUV39H1 and DNMT proteins can physically interact (references below). We have expanded the discussions to include this point.
- Fuks F, Hurd PJ, Deplus R, Kouzarides T. The DNA methyltransferases associate with HP1 and the SUV39H1 histone methyltransferase. Nucleic Acids Research. 2003;31: 2305–2312. doi:10.1093/nar/gkg332
- Yang Y, Liu R, Qiu R, Zheng Y, Huang W, Hu H, et al. CRL4B promotes tumorigenesis by coordinating with SUV39H1/HP1/DNMT3A in DNA methylation-based epigenetic silencing. Oncogene. 2015;34: 104–118. doi:10.1038/onc.2013.522
2. Based on the simulation results shown in Figure 3, the authors claimed the presence of bistability in this model. The two-state model is often characterized by a clear threshold response with deterministic bistability; however, this characteristic does not seem to be obvious, as illustrated in Figure 3A. In other words, I found that hysteresis does not clearly appear in this model, making me wonder whether or not the relation between methylated and unmethylated would be a transient without a clear threshold? Please comment on it.
> We would like to clarify this. The threshold parameter values are where the desilenced and silenced branches meets (blue and red curves). Hysteresis may be observed is a parameter changes across such threshold values. We have elaborated this in section 2.3 as “Hysteresis may be observed when the cooperative methylation rate varies continuously, and where the silenced and desilenced branches meet represent the threshold value for bistability.”
3. In the Materials and Methods, 4.4. QSSA (p. 10), it is not clear to me why the authors consider that RNA is in equilibrium with respect to M and CN at any given time if they assume that RNA saturates much faster than M and CN. Please comment on it.
> Thanks for pointing this out. The time scale of RNA-associated processes is in the order of minutes whereas methylation (M) and copy number (CN) changes over hours/days and over generations, respectively. Therefore, the changes of RNA is much faster than M and CN dynamics.We have included the following in section 4.4: “The time scales of these dynamics are different with minutes for the transcription and RNA lifetime, hours to days for the methylation, and several days for the copy number changes. Therefore, we can assume that the dynamics of RNA happen while M and CN holds constant.”.
4. In the Introduction (line 97, p. 3), the authors claimed that the state of a repeat in the next generation depends on its state in the previous generation. However, the simulation results performed in the spatial model do not seem to support this conclusion, given that, overall, methylated repeats are dominant across testing generations, except generation 40. If the probability of conversion between methylated and unmethylated stages is stochastically determined, as mentioned in the Materials and Methods (then I assume it is a random walk modeling), shouldn’t each state (or generation) be independent? Can the authors explain more about the dependency mentioned between two generations?
> The spatial model is Markovian, where the next state only depends on the current state. However, each generation is not independent from one another. Especially the cooperative methylation is local, and it requires knowing the states of neighboring states. Depending on the number of methylated neighboring repeats, the cooperative methylation rate is constant or zero (calculated at each repeat separately). Therefore, N+1 generation states is determined by N generation states. The analogy would be the random walk, and the N+1 random walks is left or right from N random walks. We have added this information to Section 2.6.
5. In the Materials and Methods, 4.6. (p. 33), is it true that U can convert to M through “demethylation’? Or “methylation”?
> Thank you for pointing this out. You are correct that the methylation is U to M. We have modified the text to correct this.
Minor concerns
1. Figure 2: It is suggested that the authors keep the same color codes for representing the parameters possessing high or low values, i.e., the color codes used in panel C are opposite from panels A and B.
> We have modified Fig. 2 to for consistency. Black dotted lines represent the reference value, and other blue lines represent cases where silencing is weakened.
Round 2
Reviewer 1 Report
Comments and Suggestions for Authors
The authors have addressed all my concerns sufficiently.
Author Response
We would like to thank you for all your suggestions.
Reviewer 2 Report
Comments and Suggestions for Authors
I appreciate the authors considering all my suggestions. The manuscript has improved, and I found only minor typos to correct:
Figure 2 – incomplete labels of y axes and legend of A shows the parameter m not f.
Line 328 – typo ‘heheterochromatin’
Author Response
Thanks for your comments.
- Figure 2 – incomplete labels of y axes and legend of A shows the parameter m not f.
> Thanks for pointing this out. We have updated the figure.
2. Line 328 – typo ‘heheterochromatin’
> We have fixed this typo.
Reviewer 3 Report
Comments and Suggestions for Authors
I appreciate the authors' effort in answering all my questions. The current version of the manuscript is well written. The results and their interpretations pertain to modeling are clear to me. I do not have further questions or comments.
Author Response

(The authors gave the same response as above.)
